



# Reconstructing the ocean's mesopelagic zone carbon budget: sensitivity and estimation of parameters associated with prokaryotic remineralization

Chloé Baumas[1]*#, Robin Fuchs[1,2]*, Marc Garel[1], Jean-Christophe Poggiale[1], Laurent Memery[3], Frédéric A.C. Le Moigne[1,3], Christian Tamburini[1]

*Both authors contributed equally

[1]Aix Marseille Univ, Université de Toulon, CNRS, IRD, MIO UM 110, Marseille, France

[2]Aix Marseille Univ, CNRS, I2M, Marseille, France

[3] LEMAR Laboratoire des Sciences de l'Environnement Marin, UMR6539, CNRS, UBO, IFREMER, IRD, Plouzané, Technopôle Brest-Iroise, France

#Corresponding author : chloe.baumas@mio.osupytheas.fr

# Abstract

Through the constant rain of sinking marine particles in the ocean, carbon (C) trapped within is exported into the water column and sequestered when reaching depths below the mesopelagic zone. Atmospheric $CO_2$ levels are thereby strongly related to the magnitude of carbon export fluxes in the mesopelagic zone. Sinking particles represent the main source of carbon and energy for mesopelagic organisms, attenuating the C export flux along the water column. Attempts to quantify the amount of C exported versus consumed by heterotrophic organisms have increased in recent decades. Yet, most of the conducted estimations have led to estimated C demands several times higher than the measured C export fluxes. The choice of parameters such as growth efficiencies or various conversion factors is known to greatly impact the resulting C budget. In parallel, field or experimental data are sorely lacking to obtain accurate values of these crucial overlooked parameters. In this study, we identify the most influential of these parameters and perform inversion of a mechanistic model. Further, we determine the optimal parameter values as the ones that best explain the observed prokaryotic respiration, the prokaryotic production, and the zooplankton respirations. The consistency of the resulting C-budget suggests that such budgets can be adequately balanced when using appropriate parameters.





**Keywords:** Biological carbon pump, Optimization methods, Carbon budget, Mesopelagic
zone, prokaryotic carbon demand, model inversion

## 1. Introduction

The biological carbon pump (BCP) is the main mechanism by which $CO_2$ is exported and stored
in the deep ocean in the long term. This ecosystem service is defined as the sum of the
biological processes that lead to carbon export from the euphotic zone into the deep ocean
(Eppley and Peterson 1979). This process exports from 5 to 20 Gt C yr$^{-1}$ in the form of
particulate organic carbon  (POC) gravitationally sinking from the sunlit ocean to the
mesopelagic zone roughly located between 200 and 1000 m (Henson et al. 2011). Therefore,
atmospheric $CO_2$ levels are strongly related to any change in carbon export into the
mesopelagic zone (Kwon et al. 2009). Five downward pathways of organic matter export to
the mesopelagic zone are defined: through phytoplankton (senescent cells, colonies, spores,
cysts), zooplankton (carcasses or fecal pellets), aggregates (marine snow of different
compositions including the two latter categories), vertical migration of zooplankton and
mixing/diffusion/advection  (Siegel et al. 2016; Le Moigne 2019).
Gravitational sinking POC supply, known as the dominant pathway, constitutes the main
organic carbon input to the mesopelagic zone (Boyd et al. 2019). Consequently, the downward
flux of organic carbon is attenuated with increasing depth as it is fragmented, metabolized and
remineralized by different biological processes until only the refractory material remains. The
majority of POC flux attenuation occurs in the mesopelagic zone (Martin et al. 1987; Marsay
et al. 2015; Fuchs et al. 2022). The remineralization of exported carbon is mainly performed
by two types of organisms: micro-organisms (mostly heterotrophic prokaryotes i.e. Bacteria
and Archaea) and zooplankton. Heterotrophic prokaryotes primarily use dissolved organic
carbon (DOC) as a source of carbon. However, some prokaryotes, colonizing particles upon
formation, undergo changes in environmental conditions during their descent, such as the
increase of the hydrostatic pressure and the variations of temperature (Tamburini et al. 2003,
2021; Baumas et al. 2021). Such particle-attached prokaryotes primarily use POC as a carbon
source. Only organic matter of size below 600 Da diffuses directly through prokaryotic
membranes, therefore attached prokaryotes produce ectoenzymes required to solubilize larger
molecules (Weiss et al. 1991). Smith et al. (1992) observed that the amount of DOC produced
by ecto-enzymatic solubilization of POC may be 10 to 100 times greater than the absorption
capacity of a cell. DOC is thereby released into the surrounding water (the so-called





solubilization). This increases the amount of DOC available for free-living prokaryotes. In
addition, several types of zooplankton are involved in marine particles: POC-feeding
detritivores (e.g. copepods), prokaryotes consumers (e.g. flagellates), and carnivores (e.g.
chaetognaths). Besides, zooplankton lose POC through excretion (moult, mucilage, urine),
fecal pellets (decomposed organic matter), and sloppy feeding. Giering et al. (2014) specify
that 30% of a particle supplied by the downward flux is fragmented by the action of the
detritivores and is transformed into suspended matter.
Given their importance regarding the BCP, all the processes described above were extensively
studied in the last decades (e.g. Alldredge and Silver 1988; Smith et al. 1992; Kiørboe et al.
2002, 2003; Kiørboe 2003; Lampitt et al. 2008; Steinberg et al. 2008; Iversen et al. 2010;
Giering et al. 2014; Koski et al. 2020 and references therein).  However, the scientific
community has struggled to reconcile the mesopelagic carbon budget with measurements and
estimates showing a biological carbon demand often greater than the amount of known organic
carbon sources (Reinthaler et al. 2006; Steinberg et al. 2008; Burd et al. 2010; Collins et al.
2015; Boyd et al. 2019). In other words, the measured export flux cannot sustain measured
metabolic demands of prokaryotes and zooplankton altogether in the mesopelagic zone, leading
to a discrepancy in C budgets.
A first explanation may lie in the choices of the boundaries of the mesopelagic zone used to
integrate fluxes and to estimate the carbon budget as investigated in Fuchs et al. (2022). Indeed
they specifically designed a method to determine from CTD-cast variables (fluorescence, $O_2$
concentration, potential temperature, salinity, and density) accurate boundaries. With their
method named RUBALIZ, they show that 90% of the POC flux attenuation occurs within new
determined boundaries which is not the case of the fixed 200-1000m often used.  Besides,
integrating prokaryotic C demand within RUBALIZ boundaries helps to reduce the
discrepancy. Other response elements may be found focusing on the carbon demand of
prokaryotes (which are responsible for the final step of the remineralization), whose estimation
is usually provided by adding rates of prokaryotic heterotrophic production (PHP) to that of
prokaryotic respiration (PR) (Burd et al. 2010). PHP rates are often measured from tritiated
leucine incorporation rates in incubations which are then multiplied by a conversion factor
Leu/Carbon (CF) (Kirchman et al. 1985). The PR is more challenging to measure (especially
in the dark ocean, (Nagata et al. 2010)) and, therefore, often estimated from measurements of
PHP and a prokaryotic growth efficiency (PGE) taken from the literature (as PR = PHP x (1-
PGE)/PGE, del Giorgio and Cole 1998). Unfortunately, *in-situ* measurements of both CF and



PGE are time-consuming and operationally complex to perform (especially for the mesopelagic
zone). In addition, such data for attached to sinking particles prokaryotic communities are
scarce since the adequate sampling devices (to specifically sample biologically intact sinking
particles) were only recently validated (Baumas et al. 2021). Besides, PHP and PR data are
usually obtained after decompression or carried out from experiments at atmospheric pressure,
being a source of misevaluation (Tamburini et al. 2013). As a result, values from the mean of
global literature compilation or theoretical values are often used as references for both CF or
PGE (Burd et al. 2010; Giering and Evans 2022) and may be far from the actual *in situ* values.
In parallel, model predictions help to estimate unmeasurable processes along with the
comparison and validation of data. The biological processes occurring in the mesopelagic zone
are not yet well constrained (see sections above). Consequently, only a few models specifically
designed to assess the fluxes governing the BCP in the mesopelagic zone exist (e.g. Tian et al.
2000; Anderson and Ryabchenko 2009; Anderson and Tang 2010; Fennel et al. 2022). For
instance, the model developed by Anderson and Tang (2010) enables the evaluation of the
remineralization of different compartments such as attached prokaryotes to sinking and
suspended particles, free-living prokaryotes and up to six trophic levels of zooplankton. This
model describes the various known biological processes involved in the BCP system. However,
the model also requires to be set up with parameters such as the PGE. For example, Anderson's
model requires 24 parameters which often present large uncertainties.
Giering et al. (2014) attempted to reconcile carbon input and biological carbon demand in the
mesopelagic zone using the Anderson and Tang (2010) model and measurements carried out
in the North Atlantic (Porcupine Abyssal Plain site, 49.0°N 16.5°W, summer 2009). They
found that prokaryotes were responsible for 70-92% of the remineralization of organic carbon.
In this study, the model results were consistent with the measurements performed *in situ*, both
showing a reconciliation of the carbon budget between 50 and 1000 m depths. Giering et al.
(2014) balanced their C-budget by using a rather low CF (CF = 0.44 kg C mol$^{-1}$) compared to
the one generally used in the literature (CF = 1.55 kg C mol$^{-1}$) and a PGE of 8% for free-living
prokaryotes and 24% for prokaryotes attached to the particles. All these values were chosen as
medians of literature values compiled from various measurement methods. Wisely choosing
these parameter is therefore crucial to determine the reconciliation or the imbalance of carbon
budget.
In this respect, we rely on model inversion methods (Tarantola 2005) to provide meaningful
estimations of parameters of interest. For a given phenomenon, inversion methods rely on a



model taking as input the parameters to be estimated and whose outputs can be compared with
*in situ* measurements. The inversion procedure thus gives the value of the parameters that best
replicate the *in situ* measurements. This type of procedure has already been used in
oceanography modeling. For instance, Saint-Béat et al. (2018) studied phytoplankton marine
food web in the Arctic and Saint-Béat et al. (2020) examined pelagic ecosystems of two
different zones in the Arctic Baffin Bay using inversion method and sensitivity analyses to
identify which biological processes impact the most the planktonic ecosystem functioning.
Here, we investigate the impact of overlooked but widely used parameters associated with the
prokaryotic remineralization (e.g. CF, PGEs) on the magnitude of the discrepancy. Our aims
are: 1) to highlight the most sensitive parameters for which the determination of an accurate
value is critical in the context of balancing of mesopelagic carbon budget; 2) to perform a
mathematical inversion method to estimate the most plausible *in situ* values of the most
sensitive parameters from a limited field dataset; 3) to discuss our results in the context of
mesopelagic carbon budget and carbon sequestration by the BCP.

# 145    2. Material & methods

## 146    2.1 *In situ* Data

Most of the data used in this study originated from the DY032 (June-July 2015) cruise at the
PAP (Porcupine Abyssal Plain) site in the North Atlantic onboard the RRS Discovery. Some
data unavailable for DY032 were estimated from a previous PAP cruise, D341 (July-August
2009). Most of the *in situ* data were compiled from already published cruise data (e.g. Giering
et al. 2014; Belcher et al. 2016; Baumas et al. 2021; Fuchs et al. 2022). Their post-treatments
to suit our study framework are described below. Additionally, we used data (ecto-enzymatic
activities along with total hydrolysable amino acids and carbohydrates, depth profile of
heterotrophic prokaryotic production and respiration under *in situ* pressure versus atmospheric)
from the PEACETIME cruise (Guieu et al. 2020) that occurred in May 2017 in the
Mediterranean Sea to illustrate some points in our discussions (see supp data).

### 157    2.1.1 Carbon fluxes

**158    a) Determination of the Active Mesopelagic zone boundaries**

Fuchs et al. (2022) introduced the "RUBALIZ" method, using CTD data, which allows the
estimation of vertical boundaries targeting the zone of the dark ocean where most of the POC





fluxes attenuation occurs. At station PAP during cruise DY032, this so-called "Active
Mesopelagic Zone" was located between 127 and 751 m.
**b) Carbon inputs**
The POC inputs to the active mesopelagic zone mainly involve the gravitational export of POC.
Gravitational input was taken from Fuchs et al. (2022) who fitted a power law Martin curve (b
of 0.84) on data obtained from 30 to 500m using Marine Snow Catcher (Belcher et al. 2016).
However, gravitational input is not the only POC input known in the literature. Recently, Boyd
et al. (2019), provided an estimation of other particle-injection pumps (PIPs) such as the mixed
layer pump, physical pump, the seasonal lipid pump or the active transport related to metazoans
migrations. At the PAP site during summer, only the eddy subduction pump, metazoans
migrations, and large-scale physical pumps were relevant to take into account. Other PIPs do
not correspond to the location and season considered in our study. From Boyd et al. (2019)
review, these three particle-injection pumps seem to represent altogether around 52% of the
gravitational export of POC. We therefore add up this proportion of POC to the purely
gravitational inputs. This yields an overall POC flux of 134 mg C m$^{-2}$ d$^{-1}$ exported into the
active mesopelagic zone. The corresponding net POC input is 117 mg C m$^{-2}$ d$^{-1}$ (that is POC
fluxes at the end - 751 m - of the active mesopelagic zone subtracted to the one at the start -
127 m - for PAP DY032).
DOC inputs are taken from Giering et al. (2014) and are considered as the sum of direct DOC
export via physical processes (advection-diffusion) and active flux from zooplankton
migrations. We estimated from their extended Data Fig. 2 that the DOC gradient below 100m
is hardly visible meaning that physical vertical DOC export is insignificant for the active
mesopelagic zone which is studied here. As a result, we set the DOC export at 3 mg C m$^{-2}$ d$^{-1}$,
which corresponds only to the active flux from zooplankton migrations from Giering et al.

185 (2014).

**c) Carbon demands**
As explained above, prokaryotic carbon demand is generally assessed by adding rates of
prokaryotic heterotrophic production (PHP) to that of prokaryotic respiration (PR). PHP of
non-sinking prokaryotes (that is free-living and attached to suspended particles prokaryotes)
are derived from leucine incorporation measurements on seawater samples and are taken from
Fuchs et al. (2022). These data did not permit the separation of the free-living from attached to
suspended particles (Baumas et al. 2021). Hence, in the sequel, we no longer make this





distinction and group both types under the term "non-sinking prokaryotes". During DY032,
Marine Snow Catchers (MSC) were deployed to separate slow and fast-sinking particles from
100L of samples (Riley et al. 2012; Baumas et al. 2021). PHP rates associated with prokaryotic
communities of fast-sinking particles were taken from Baumas et al. (2021) and slow-sinking
particles are presented here. Briefly, slow-sinking particle fractions were sampled in the 7L
base of the MSC. Samples were incubated and leucine incorporation rates were measured as
for fast-sinking particles in Baumas et al. (2021). The formula described in Baumas et al. (2021)
was then applied to normalize to 100L as particles were concentrated in 7L after 2h of
decantation and to remove the contribution of non-sinking prokaryotes which were primarily
in this compartment around slow-sinking particles of interest. Total sinking prokaryotes PHP
rates were obtained by adding both fast-sinking and slow-sinking prokaryotes PHP rates. In
addition, we were able to use the respiration rates of prokaryotes attached to fast-sinking
particles obtained by Belcher et al. (2016). For each depth (30-500m) the mean total $O_2$
consumption per particle in nmol agg$^{-1}$d$^{-1}$ was converted to mg C m$^{-3}$ d$^{-1}$ (assuming a respiration
quotient RQ($CO_2/O_2$) = 1) by multiplying by the total number of particles (i.e. fecal pellets +
phytoplanktonic aggregates) and dividing by 95L which is the volume of the MSC used (Riley
et al. 2012). It is also important to note that PR for slow-sinking particles is missing. Thus,
when we mention the respiration of sinking prokaryotes, only attached to fast-sinking
prokaryotes are taken into account which certainly underestimates the respiration used. All
prokaryotic carbon demand (PHPs and PRs) estimates were integrated within RUBALIZ
boundaries (i.e. 127m - 751m). Non-sinking prokaryotes PHP rates were integrated using a
piecewise model with a single node on the log-data as described in Fuchs et al. (2022). Sinking
prokaryotes PHP rates were integrated using power law. Sinking PR were integrated using
trapeze because data are only available until 500m and without any *a priori* on the curve shape,
this method is certainly the most conservative.
Zooplankton activities are known to be related to POC concentration (Steinberg et al. 2008).
Zooplankton respiration data were available only for the cruise D341 when the net POC input
into the active mesopelagic layer was 59 mg C m$^{-2}$ d$^{-1}$ (including PIPs) instead of 134 mg C m$^{-2}$ d$^{-1}$
$^{-2}$ d$^{-1}$ for DY032 (see above). For D341, zooplankton respiration integrated within the active
mesopelagic zone (135-726m, Fuchs et al. 2022) was 9 mg C m$^{-2}$ d$^{-1}$. Zooplankton respiration
was integrated using a power law as in Giering et al. (2014). Zooplankton respiration data are
missing for DY032, thus we consider this quantity as a percentage of the POC input that we
calculate from the D341 data set, i.e. 14.67%. The zooplankton respiration value used here is
therefore 17 mg C m$^{-2}$ d$^{-1}$.



*Table 1: Fluxes and their associated values used in this study. Anderson & Tang model's terms*

*(Anderson and Tang 2010) corresponding to these fluxes are also shown. Values are integrated*

*between 127 and 751m which are boundaries of the active mesopelagic zone defined by* Fuchs

*et al. (2022). POC and DOC refer respectively to Particulate and Dissolved Organic Carbon,*

*PHP to Prokaryotic Heterotrophic Production, and PR to Prokaryotic Respiration.*

| Name | Anderson and Tang's Model term correspondence | Values | units | sources |
|---|---|---|---|---|
| Net POC input | $D1ex$ | 117 | mg C m$^{-2}$ d$^{-1}$ | Belcher et al. (2016); Boyd et al. (2019) |
| DOC input | $DOCex$ | 3 | mg C m$^{-2}$ d$^{-1}$ | Giering et al. (2014) |
| Non-sinking prokaryotes PHP | $F_{BFL}+ F_{BAD2}$ | 1.10E+07 | pmol Leu m$^{-2}$ d$^{-1}$ | Baumas et al. (2021) |
| Sinking prokaryotes PHP | $F_{BAD1}$ | 1.02E+06 | pmol Leu m$^{-2}$ d$^{-1}$ | Baumas et al. (2021) |
| Sinking prokaryotes PR | $R_{BAD1}$ | 19 | mg C m$^{-2}$ d$^{-1}$ | Adapted from Belcher et al. (2016) |
| Zooplankton respiration | $R_{VA}+R_{VFL}+R_H+R_{Z1:6}$ | 17 | mg C m$^{-2}$ d$^{-1}$ | Adapted from Giering et al. (2014) |

## 2.2 Mathematical methods

### 2.2.1 Parameter estimation

The scope of our study is to estimate *in situ* parameters by inverting the model introduced by

Anderson and Tang (2010), adapted by Giering et al. (2014). We do not intend to present the

model in details here. The details of the equations constituting the version of the model used

can be found in the original paper (Anderson and Tang 2010), in the R code available at

https://github.com/RobeeF/InverseCarbonBudgetEstim and the specific terms related to

variables used are reported in Table 1. The model is calibrated by choosing the set of input



parameters that yields the best fit between the model output and the data. As the model outputs 85 outfluxes, we used a subset of four measurable outfluxes to calibrate the model: the PHP of non-sinking prokaryotes, the PHP of sinking prokaryotes, the PR of sinking prokaryotes and the respiration of zooplankton. These fluxes have been chosen because of their near direct correspondence with outputs of the model linked to the C demand of all groups (sinking prokaryotes, non-sinking prokaryotes, detritivores, bacterivores, and carnivores).

Similarly, the model relies on 24 input parameters (Table S1), which makes the parameter space of significant size and therefore challenging to explore. As such, we first determine the set of parameters that have the largest impact on the output of the model. Then for these parameters, the values that give the best fit between the data and the solution given by the model are determined.

**a) Sensitivity of the model to its inputs**

In order to reduce the size of the input parameter space, Sobol Indices (Sobol 1993) were used to determine the most influential parameters. These indices enable quantification of the share of the variation of the output that can be imputed to each input parameter.

In essence, the first-order Sobol indices account for the direct influence of an input variable on the output. However, first-order Sobol indices neglect the interactions existing between this input variable and the other input variables. As such, in addition to the first-order Sobol Indices, we used the total Sobol indices introduced by Homma and Saltelli (1996) which encompass both the direct effect of a parameter and also its interactions with the other parameters.

First-order and total Sobol indices were computed to quantify the influence of each parameter over each of the four outfluxes. Only the parameters which had significant Sobol indices (i.e. Sobol indices > 0.20) for at least one outflux were kept.

**b) Estimation of the parameters**

The parameters which had no substantial effects on the output of the model were set to the values indicated by Anderson and Tang (2010) and Giering et al. (2014) and given in Appendix (Table S1). The other parameters were estimated by minimizing the distance existing between the four outfluxes predicted by the model and their *in situ* measured counterpart. The distance chosen here is a standardized Euclidean distance:






$$\sum_{i=1}^{4}\left(\frac{outflux_{obs,i} - outflux_{model,i}}{outflux_{obs,i}}\right)^2 \qquad (1)$$

where $outflux_{obs,i}$ is the i-th measured flux and $outflux_{model,i}$ its modeled counterpart. The
optimization method used is the Nelder-Mead algorithm (Nelder and Mead 1965): if the
function to minimize depends on N variables (the number of input parameters here), a simplex
constituted by N + 1 points is defined. The coordinates of the simplex are updated in turn so
that the simplex vertices get closer to the local minimum. Even if this method gives little
theoretical guarantees of convergence, it has proven to work well in practice (Lagarias et al.
1998) and has the advantage that it does not require computing the gradient of each outflux
with respect to each input parameter.
As the model takes 24 inputs and outputs 85 fluxes, concerns might be raised about the
uniqueness of the solution found to minimize the term (1). To make the model identifiable, the
number of input parameters to estimate is limited to the number of output fluxes available, here
four. In this respect, the CFs have been fixed to 0.5 kg C mol Leu$^{-1}$. This value, contrary to the
previously classically used value of 1.55 kg C mol Leu$^{-1}$ (Simon and Azam 1989; Nagata et al.
2010), was determined by Giering and Evans (2022) as the median value of 15 studies
conducted in the mesopelagic zone. Doing so, we limit the number of free parameters to be
estimated to four so that the model remains identifiable. The model is mostly linear and our
experiments have shown the solution to be unique and independent of the initial values taken.
The codes and data to reproduce the results are available at
https://github.com/RobeeF/InverseCarbonBudgetEstim

# 291 **3. Results**

## 292 **3.1 Most sensitive parameters**

Using Sobol indices, we identified the most sensitive parameters from the 24 of the Anderson
and Tang (2010) model on the 4 fluxes outputs of the model for which we have the measured
counterpart (i.e. PHP and PR of sinking prokaryotes, PHP of non-sinking prokaryotes and
respiration of zooplankton). All parameter definitions are given in Table S1. For the outflux
"PHP of non-sinking prokaryotes", only the $PGE_{non-sinking}$ appears to be sensitive with a Sobol
index of 0.68 meaning that it explains 68% of the variance (Table 2). Fluxes related to sinking
prokaryotes, i.e. their PHP and their PR, appear to be highly influenced both by $\Psi$, $\boldsymbol{\alpha}$, and





PGE$_{sinking}$ with indices of 0.22 and 0.23 for $\Psi$, 0.24 and 0.24 for **α** and 0.27, 0.25 for PGE$_{sinking}$
respectively. Surprisingly, zooplankton respiration is more impacted by the PGE$_{non-sinking}$
(Sobol index of 0.52) than proper zooplankton parameters. All other parameters exhibit Sobol
indices below 1%. Total Sobol indices, indicating the part of the variance of fluxes due to the
parameter alone and in interaction with the others, were similar to the first-order indices,
suggesting no interactions of parameters regarding the variance of fluxes. This sensitivity
analysis enabled the identification of $\Psi$, **α**, and both PGEs as the most influential parameters,
suggesting that their values should be set with particular care. Especially for the PGE$_{non-sinking}$
which can be responsible for more than 50% of the variance of PHP$_{non-sinking}$ and zooplankton
respiration. PGEs are growth efficiencies defined as the amount of new prokaryotic biomass
produced per unit of organic C substrate assimilated and is a way to relate PHP and PR (del
Giorgio and Cole 1998). $\Psi$ corresponds to the percentage of POC consumed by prokaryotes
and **α** to the fraction of hydrolyzed POC which is lost into the surrounding water, i.e. not
assimilated by sinking prokaryotes that hydrolyzed it.
*Table 2: First-order Sobol indices for the parameters of the model by Anderson and Tang*
*(2010). The definition of each parameter can be found in Table S1. Significant Sobol indices*
*(>0.2) are shown in red. PHP and PR respectively refer to Prokaryotic Heterotrophic*
*Production and to Prokaryotic Respiration.*

| | ψ | PGE sinking | PGE non-sinking | α | Φv | βv | Kv | Φv | βv | Kv | Φz | βz | λz | Kz | Φh | βh | λh | Kh | ζ | ζ2 |
|---|---|---|---|---|---|---|---|---|---|---|---|---|---|---|---|---|---|---|---|---|
| Non-sinking prokaryotes PHP | <0.01 | 0.021 | **0.681** | 0.01 | <0.01 | <0.01 | <0.01 | 0.014 | <0.01 | 0.011 | -0.012 | <0.01 | <0.01 | 0.015 | <0.01 | <0.01 | <0.01 | <0.01 | <0.01 | <0.01 |
| Sinking prokaryotes PHP | **0.222** | **0.24** | <0.01 | **0.265** | <0.01 | <0.01 | <0.01 | <0.01 | 0.011 | <0.01 | <0.01 | <0.01 | <0.01 | <0.01 | 0.011 | <0.01 | <0.01 | -0.012 | <0.01 | -0.011 |
| Sinking prokaryotes PR | **0.225** | **0.243** | <0.01 | **0.252** | -0.019 | <0.01 | <0.01 | <0.01 | -0.011 | <0.01 | <0.01 | <0.01 | 0.012 | <0.01 | <0.01 | <0.01 | <0.01 | <0.01 | <0.01 | <0.01 |
| Zooplankton respiration | <0.01 | 0.023 | **0.507** | <0.01 | <0.01 | 0.014 | <0.01 | 0.064 | 0.027 | 0.041 | <0.01 | <0.01 | <0.01 | <0.01 | <0.01 | <0.01 | 0.028 | <0.01 | <0.01 | <0.01 |


**3.2 Model inversion**






The optimization method, described in the material and method section, enabled the
determination of the 4 parameters identified as sensitive above: $\Psi$, $\alpha$, $PGE_{sinking}$, and $PGE_{non-}$
$_{sinking}$ in the case study of PAP DY032. Table 3 reports the combination found by model
inversion. By construction of the procedure (e.g. same number of input and output), the solution
is unique, explaining why no confidence intervals are reported. The errors between the four
fluxes generated by the model and their measured counterparts were less than 1%, far lower
than potential measurement errors. The zooplankton flux was the best matched, followed by
the PR of the sinking prokaryotes, the PHP of the non-sinking prokaryotes, and of the sinking
prokaryotes.

*Table 3: Estimation of the parameters* $\Psi$, $\alpha$, $PGE_{sinking}$ *and* $PGE_{non-sinking}$ *obtained by inversion*
*of the model by Anderson and Tang (2010). As the model was made identifiable, the solutions*
*are unique, explaining the absence of confidence intervals. The remaining differences between*
*the model outfluxes deriving from the estimated input values and the actual in situ*
*measurements are referred to as "Errors" and are expressed in percentage. PHP, PR, and ZR*
*respectively stand for Prokaryotic Heterotrophic Production, to Prokaryotic Respiration and*
*to Zooplankton Respiration.*

| Estimations | | | | Errors | | | |
|---|---|---|---|---|---|---|---|
| $\Psi$ | $\alpha$ | $PGE_{sinking}$ | $PGE_{non-sinking}$ | $PHP_{non-sinking}$ | $PHP_{sinking}$ | $PR_{sinking}$ | ZR |
| 0.675 | 0.777 | 0.026 | 0.087 | -0.487% | 0.524% | 0.184% | -0.05% |


## 3.3 C budget


The two PGEs presented above along with CF of 0.5 kg C mol Leu$^{-1}$ were applied to leucine-
incorporation rates measurements to build the corresponding active mesopelagic C budget. The
resulting C budget was compared with two other C budgets calculated with different sets of
parameters. The three active mesopelagic zone C budgets resulting from DY032 measurements
or estimation are represented in Fig. 1 with the budget (1) obtained with the classical CF value
of 1.55 kg C mol Leu$^{-1}$ and median literature values for PGEs, i.e. 0.07 for $PGE_{non-sinking}$
(Arístegui et al. 2005; Reinthaler et al. 2006; Baltar et al. 2010; Collins et al. 2015) and 0.02
for $PGE_{sinking}$ (Collins et al. 2015); the budget (2) obtained with the parameter values from
Giering et al. (2014) who reconcile C budget, i.e. CF of 0.44 kg C mol Leu$^{-1}$, $PGE_{non-sinking}$ of



0.07, and PGE$_{sinking}$ of 0.24 and the budget (3) obtained with a CF, PGE$_{sinking}$ and PGE$_{non\text{-}sinking}$
of 0.5, 0.026 and 0.087, respectively, determined in this study. The combination yielding to the
largest discrepancy is the budget (1) (Fig. 1) (discrepancy of -194 mg C m$^{-2}$ d$^{-1}$). The C input
seems to support the zooplankton respiration and total C demand of sinking prokaryotes but
not the one of non-sinking prokaryotes especially due to their PR of 218 mg C m$^{-2}$ d$^{-1}$.
Combination of budget (2) and (3) presented both an excess of C (60 and 40 mg C m$^{-2}$ d$^{-1}$
respectively) compared to the biological C demand. These two differ mainly on the PR of
sinking prokaryotes which is negligible in combination (2) but which is the second largest flux
in the C demand in our study. In all cases, the C demand of non-sinking prokaryotes accounts
for most of the total C demand.

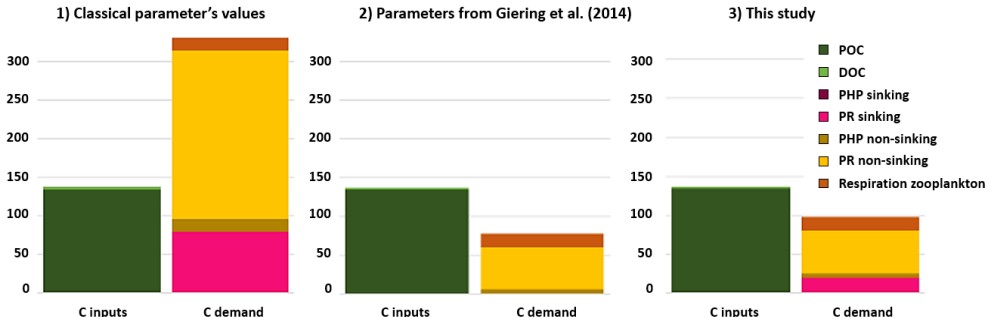


*Figure 1: Carbon budget for the active mesopelagic zone estimation resulting from DY032*
*measurements or estimation and on which different combination of CF (1.55, 0.44 and 0.5 respectively*
*for budget 1) 2) and 3)) , PGE$_{sinking}$ (0.02, 0.24 and 0.026 respectively for budget 1) 2) and 3)) and*
*PGE$_{non\text{-}sinking}$ (0.07, 0.08 and 0.087 respectively for budget 1) 2) and 3)) were applied on leucine*
*incorporation rates of sinking and non-sinking prokaryotes. See Fig. S1 for value  details.*

# 4.Discussion:

As stated in the introduction, the scientific community has struggled to reconcile the
mesopelagic carbon budget with measurements and estimates showing a carbon demand often
greater than the amount of known organic C sources (e.g. Reinthaler et al. 2006; Steinberg et
al. 2008; Burd et al. 2010; Collins et al. 2015; Boyd et al. 2019). Building C budget involves a
plethora of parameters whose impacts are overlooked and often neglected, mainly because
neither their ideal values nor their underlying mechanism in the water column across space and
time are clearly understood. The scientific community is concerned about this issue (e.g. Burd
et al. 2010; Giering and Evans 2022), but in the absence of a better option and in an attempt to
encourage comparisons, the same parameter values are universally used. A first step towards



this direction was conducted thanks to the RUBALIZ method (Fuchs et al. 2022) which
precisely determines the vertical location of the "active mesopelagic zone" and thereby
estimates the boundaries between which to integrate C fluxes. In the current study, we pursue
this investigation and combine measurements with modeling approaches to investigate the role
of sensitive parameters related to the remineralization of POC in the mesopelagic zone.

## 4.1 Optimization method: Consistency of parameters estimated

The Anderson and Tang (2010) model takes as inputs the measured C inputs as well as 24
parameters related to the activity of organisms such as sinking prokaryotes, non-sinking
prokaryotes, zooplankton detritivores, bacterivores, and carnivores. Among the 24 parameters,
four have been found to be particularly sensitive in assessing the carbon demands of the various
groups: $\Psi$ (percentage of particle consumption by prokaryotes), $\alpha$ (percentage of C hydrolyzed
released in surrounding water), $PGE_{non-sinking}$ and $PGE_{sinking}$ (growth efficiencies of sinking and
non-sinking prokaryotes). It is interesting to note that zooplankton respiration (which is the
sum of detritivores, bacterivores and carnivores respiration) is mostly sensitive to one
parameter: $PGE_{non-sinking}$ but not to a parameter specific to zooplankton. This counter-intuitive
result suggests a strong synergy between the two model compartments. At this point, it is
challenging to establish whether this is the outcome of a complex ecological process or a model
artifact.

In the model, the consumption of particles is done by two groups: prokaryotes ($\Psi$) and
detritivores (1-$\Psi$). It can be estimated by taking the average ratio between PHP and ZR.
Anderson and Ryabchenko (2009) estimated $\Psi$ using calculations of POC consumption by
prokaryotes and zooplanktons between 150 and 1000m performed by Steinberg et al. (2008) in
the Pacific. Following this, they set $\Psi$ at 0.76. The inversion of the Anderson and Tang model
(2010) leads to a well-identified solution of $\Psi$, i.e. 0.67 in the case of PAP DY032 cruise. This
value is in line with the one used by Anderson and Tang (2010). However, data are lacking to
compare and explore variations of $\Psi$ value across seasons, locations or depths. In the model, $\Psi$
participates in the repartition of POC input between prokaryotes and detritivores. Whether for
modeling purposes to determine $\Psi$ or to build a C-budget without a model, PHP and ZR are
required. It remains too rare to have both together and more future efforts should be devoted to
get PHP and ZR concomitantly.

Beyond $\Psi$, according to Sobol indices, $\alpha$ is the second parameter of interest. Amino acids and
sugar are major components of POC, constituting between 40 to 70% of POC in the



mesopelagic zone (Wakeham et al. 1997). When prokaryotes consume POC using hydrolytic
enzymes, a major fraction of the hydrolyzed C is lost to the surrounding environment as DOC
(Smith et al. 1992; Vetter et al. 1998). This loss is represented by $\alpha$ and is very difficult to
quantify accurately. Two major experiments, focused on amino acid hydrolysis, aimed to
determine such losses: Smith et al. (1992) and Grossart and Ploug (2001). Smith et al. (1992)
sampled particles at 25m and showed that 97% of particulate combined amino acids are
released in the surrounding water. Later, Grossart and Ploug (2001) using aggregates from
phytoplankton cultures show a loss of POC of 74%. Relying on these two studies, Anderson
and Tang (2010) followed by Giering et al. (2014) consider that the value should be lower than
that of a fresh detritus and choose a conservative value of 0.5. In the case of these two
experiments, only the amino acids are considered and the experiments were conducted under
laboratory-controlled settings. Conversely, we used unpublished data from PEACETIME
cruise (see methods details in supp. data) of *in situ* hydrolysis rates of aminopeptidase and $\beta$-
glucosydase from sinking prokaryotes (which hydrolyze amino acids and sugar, respectively)
that we were able to convert into hydrolyzed carbon fluxes (see measurements and calculation
details in supp. data). Unfortunately, total hydrolyzed C fluxes were most of the time below
the C demand of the sinking prokaryotes which is unrealistic and probably due to the low
amount of POC (sinking POC concentration of <1 mg L$^{-1}$ in the sinking fraction) resulting in
insufficient sinking prokaryotes abundance to detect their activity by volume. However, when
total hydrolyzed C fluxes were superior to PHP$_{sinking}$ (indicating that some hydrolyzed C is not
assimilated and is released), $\alpha$ was estimated between 0.19 and 0.79 with a mean of 0.41±0.24
and seems to decrease with depth (see calculations details in supp data). This could confirm
Grossart and Ploug's (2001) work showing that the older a detritus is, the less enzymatic
activity there is and therefore the less amino acid loss. Even if $\alpha$ is not measurable easily, this
parameter is identified at 0.78 by the inversion method during a post-bloom period at the PAP
site. This value is consistent with Smith et al. (1992) and Grossart and Ploug (2001) evidencing
high $\alpha$ for surface aggregates (0.97) with laboratory-made phytoplankton aggregates (0.74), or
with our calculations for the Mediterranean Sea (0.41±0.24), an oligotrophic region. This
suggests that the optimization method is a relevant alternative to determine $\alpha$. In addition, $\alpha$
corresponds to a release of C in the surrounding water. Regarding the model, the C demand of
free-living prokaryotes matches the hydrolyzed C released which constitutes their main C
sources. The relationship between enzymatic activities and heterotrophic production of free-
living prokaryotes is well documented in the deep-sea ocean (Cho and Azam 1988; Smith et
al. 1992; Hoppe and Ullrich 1999; Tamburini et al. 2002, 2003; Nagata et al. 2010). Total C
demand of non-sinking prokaryotes is challenging to measure due to the diversity of existing



methods, especially the PR (e.g. Table S2), which leads to an incredibly wide range of
estimated values. Subsequently, identifying **α** via the optimization method could help to avoid
these conflicting PR measurements.

The last two sensitive parameters according to Sobol indices were $PGE_{non-sinking}$ and $PGE_{sinking}$.
A wide range of $PGE_{non-sinking}$ has been estimated using $PHP_{non-sinking}$ and $PR_{non-sinking}$ in the open
ocean (e.g. Sherry et al. 1999; Lemée et al. 2002; Carlson et al. 2004; Arístegui et al. 2005;
Reinthaler et al. 2006; Baltar et al. 2009, 2010; Collins et al. 2015). Overall it varies from 0.001
to 0.64 (Collins et al. (2015) and Sherry et al. (1999), respectively). However, these values
were produced from different protocols for the PHP (changes in biomass, thymidine or leucine
incorporation, each with its own conversion factors and/or constants) and for the PR methods
(by ETS measurements, micro-winkler titration, changes in dissolved $O_2$, or using optodes
sensors spots, see Table S2) and correspond to various locations, seasons and depths. These are
all valid reasons that can potentially explain the stark contrast in the values reported. If one
focuses only on the mesopelagic zone in the North Atlantic, the median is 0.07 (Arístegui et al.
2005; Reinthaler et al. 2006; Baltar et al. 2010; Collins et al. 2015). The optimization method
yielded to a value of 0.087 and therefore produces very consistent results for a post-bloom
period at the PAP site. Concerning $PGE_{sinking}$, too few values are available. To our knowledge,
only Collins et al. (2015) provided *in situ* values associated with sinking prokaryotes (from
0.01 to 0.03) at 150m. This is the only comparison we have, and our value of 0.026 matches
this order of magnitude. To provide further comparison, the DY032 data before integration
allows us to calculate a $PGE_{sinking}$ (using $PGE_{sinking} = PHP_{sinking}/(PHP_{sinking}+PR_{sinking})$ from del
Giorgio and Cole 1998) per $PR_{sinking}$ and $PHP_{sinking}$ of sinking prokaryotes points performed at
the same depth. This led to a variation from 0.033 at 70m to 0.0013 at 500m. Although the lack
of datapoints deeper than 500m and the low number of points forces us to stay cautious about
these estimates, it may indicate that $PGE_{sinking}$ is not constant throughout the mesopelagic zone
and decreases with depth. Constraining conditions due to the increase of hydrostatic pressure
and decrease in temperature experienced by prokaryotes attached to sinking particles could
explain this decrease in $PGE_{sinking}$ (Stief et al. 2021; Tamburini et al. 2021). Under highly
constrained conditions, Russell and Cook (1995) explained that maintaining respiration at the
highest possible rate would allow the supply of active membrane transporters which are vital
to the cell. This implies a low but optimal PGE (Westerhoff et al. 1983) which could thus
decrease with depth and time as the POC becomes less labile (Grossart and Ploug 2000). On
the contrary, the Anderson and Tang (2010) model, and the associated model inversion
presented here, is built so that the mesopelagic zone is considered as one homogeneous entity.





Explicitly, specifying depth-dependent $PGE_{sinking}$ in the mesopelagic zone could lead to more
realistic modeling, but would entail a non-negligible additional model complexity.

It is worth noting that the $PGE_{sinking}$ and $PGE_{non-sinking}$ estimated here rely on a leucine-to-carbon
Conversion Factor (CF) of 0.5 kg C mol Leu$^{-1}$. This value comes from the median of 15 values
obtained on the free-living prokaryotes of the mesopelagic zone (between 300 to 1000m),
which do not sink and are adapted to their place in the water column (Giering and Evans 2022).
However, to our knowledge, there are no such values measured for the specific case of sinking
prokaryotes. The latter are surface prokaryotes that have attached to the particles and will
experience changes in conditions (e.g. pressure, temperature) linked to their sink (Baumas et
al. 2021; Tamburini et al. 2021). The CF depends, among other things, on the leucine fraction
in the proteins and the cellular carbon/protein ratio (Kirchman and Ducklow 1993). It is known
that stresses can affect the incorporation of leucine into proteins and general protein production
(e.g. Young 1968; Welch et al. 1993) and that these parameters can vary with prokaryotic
diversity, especially between bacteria and archaea (Bogatyreva et al. 2006). Stresses occur
during the descent throughout the water column and sinking prokaryotes experienced a drastic
decrease in diversity following the sink at PAP DY032 (Baumas et al. 2021; Tamburini et al.
2021). We can therefore easily imagine that the CF for sinking prokaryotes could be impacted.
Despite this, without having further data, we applied the same CF on sinking as the 0.5
recommended by Giering and Evans (2022) for non-sinking prokaryotes and the results were
consistent.

## 503  4.2 Influence on mesopelagic C Budget

As stated in the introduction, mesopelagic C budgets are constructed by applying a CF and a
PGE on leucine incorporation rates data to assess prokaryotic C demand. In Fig. 1, we applied
three different combinations of CFs and PGEs to the same data. The combination using
conventional CF of 1.55 kg C mol Leu$^{-1}$, $PGE_{non-sinking}$ of 0.07, and $PGE_{sinking}$ of 0.02 led to an aberrant
discrepancy such that more than the entire C pool would be remineralized in the active
mesopelagic zone and that there would be no source of C to sustain deeper zone life nor
sequestration by the BCP. As stated above, this was a recurrent issue in the field (Reinthaler et
al. 2006; Steinberg et al. 2008; Burd et al. 2010; Collins et al. 2015; Boyd et al. 2019) with the
exception of Giering et al. (2014) who reconcile the C budget of the mesopelagic zone.
Although Giering et al. (2014) did not take into account sinking prokaryotes from *in situ* data
point of view, their results were mainly due to the difference in CF used, i.e. 0.44 kg C mol Leu$^{-}$





[1]. However, from a model point of view, the main difference between C budgets estimated
using Giering et al. (2014) parameters and those determined by our optimization method is due
to the 10-fold difference between $PGE_{sinking}$ used. Giering et al. (2014) used 0.24 which is the
mean of a 14 days incubation experiment during which PGE varied from 0.45 in the first 3 days
to 0.04 at the end for riverine aggregates (Grossart and Ploug 2000). Despite the fact that
$PGE_{sinking}$ data are very scarce, riverine values of 0.24 seem highly unlikely and inappropriate
to mesopelagic sinking prokaryotes compared to what is known in marine environments (e.g.
Collins et al. 2015). Indeed, if we consider that enzymes account for a large proportion of the
proteins produced by cells (see above) the $PGE_{sinking}$ must be low due to the high metabolic
cost of their production (Grossart and Ploug 2000). Finally, the C budget built from a
combination of CFs of 0.5 kg C mol Leu$^{-1}$ and PGEs revealed by our optimization method seems
the most reasonable option (from the three budgets built, Fig. 1) with an excess of C input of
40 mg C m$^{-2}$ d$^{-1}$. In this case, PGEs were determined by the model, which in addition to PHP
and PR of sinking and non-sinking prokaryotes and zooplankton respiration, also accounts for
the production of zooplankton biomass into calculations. We do not have measurements or
estimates for the production of zooplankton biomass but based on the model, this biomass
production is 11 mg C m$^{-2}$ d$^{-1}$. Adding this value to the C demand implies a leftover of 29 mg
C m$^{-2}$ d$^{-1}$ that is not used and is exported under the active mesopelagic zone via gravitational
sinking POC. This value is in accordance with the POC flux estimated from measures at 751m
(thus at the exit of our zone): 17 mg C m$^{-2}$ d$^{-1}$. Being aware of the biases that may exist in the
fluxes used as well as in the construction of the model itself, our optimization method enables
the determination of realistic values of parameters and thus constructing robust C budgets. As
far as we know, the combination of field measurements (using consistently defined integration
depths, such as RUBALIZ (Fuchs et al. 2022) with the use of optimization method on the
Anderson & Tang model has led to the most complete and realistic mesopelagic carbon budget.

## 4.3 Model: reliability and potential biases

The Anderson and Tang model (Anderson and Tang 2010) was originally parametrized with
24 input parameters and 85 output fluxes, and is hence by definition an underdetermined model
as the number of outputs is higher than the number of inputs. To make the model identifiable,
i.e. obtaining unique solutions for each parameter value, the number of parameters allowed to
vary, namely: $\Psi$, $\alpha$, $PGE_{non-sinking}$, and $PGE_{sinking}$, was restricted to the number of measurable
outputs (here four, $PHP_{sinking}$, $PR_{sinking}$, $PHP_{non-sinking}$, and zooplankton respiration).
Measurement errors (e.g. measurement device errors, *in situ* variabilities, errors due to





integration methods) are typically challenging to characterize. Furthermore, even if these four
outfluxes well describe the prokaryotic and zooplankton compartment fluxes, one may wonder
about the sensitivity of the results to the fact that a given outflux is not available or estimated
with error.

As a result, we have tested two settings: a model inversion without the zooplankton respiration
flux (using only three fluxes) and a second setting where the PGEs were estimated from the
leucine incorporation rate using freely varying CFs, i.e. with CFs no more fixed at 0.5 as a
value. The results are reported in Table S3 and S4. Not using the zooplankton flux to inverse
the model mechanically adds some variability to the estimation results, especially concerning
$\Psi$, $\alpha$, and $PGE_{non\text{-}sinking}$, in decreasing order of variability (Table S3). The $PGE_{sinking}$ was not
affected as its confidence interval length was inferior to $10^{-7}$: this underlines the very limited
influence between the zooplankton and sinking prokaryote compartments in the model
(contrary to the zooplankton and non-sinking prokaryote compartments). Yet, the difference
between the four-flux and three-flux parameter estimations was negligible (<1% variation for
each estimate), highlighting the robustness of the estimates to the potential unavailability of
the zooplankton respiration. On the contrary, as made visible in Table S4, not fixing the CFs
to estimate the PGEs created more variations in the PGE estimations, while the estimations of
$\Psi$ and $\alpha$ changed by less than 5% with respect to Table 2 estimations. The PGEs of the attached
and free-living parameters get significantly closer to their fixed boundaries (10%), while the
CFs rise, especially the CF of the attached particles (=1.865). Similarly, if PGEs are no longer
bounded, the estimates of PGEs (0.173 for attached prokaryotes and 0.226 for free-living
prokaryotes) and CFs (3.927 for attached prokaryotes and 1.526 for free-living prokaryotes)
become unrealistic. This can be explained by the fact that the PGEs and CFs play similar roles
in the current formulation of the model. Hence, without additional fluxes ensuring full model
identifiability, one of these two types of quantities needs to be fixed to estimate the other.

In addition to these sensitivity analyses, an uncertainty analysis has been run by simulating
errors in the measurements of the POC, DOC and the four output fluxes (see Table S5 in supp.
data). Simulating errors from -10% to 10% for each flux, the estimation of the four parameters
of interest were lowly affected: 1%, 2%, 3% and 1% on average for the $\Psi$, $PGE_{sinking}$, $PGE_{non\text{-}}$
$_{sinking}$ and $\alpha$, respectively. The $PGE_{non\text{-}sinking}$ was mostly sensitive to measurement errors of POC
flux, DOC flux and $PHP_{non\text{-}sinking}$ (generating variations of 6%, 5% and 5%, respectively).
Similarly, the $PGE_{sinking}$ was logically mostly sensitive to errors in the $PHP_{sinking}$ and $PR_{sinking}$



(generating variations of 6% for both). For the measurement errors, the generated variations all
remained under 3% which is reassuring concerning the stability of the estimation.

Finally, the last potential source of estimation bias results from the assumed stationarity
hypothesis of the mesopelagic system. For logistical and technical reasons, measurements and
sampling between the upper and lower boundary of the mesopelagic zone are typically
performed simultaneously. The stationarity assumption is thus a natural foundation ground
upon interpretations and models. However, there is a temporal delay in flux variations between
the upper layer and lower measurements (Giering et al. 2017; Stange et al. 2017). This delay
depends on the particles sinking speed typically ranging from 2 to 1500 m d$^{-1}$ (Alldredge and
Silver 1988; Armstrong et al. 2002; Trull et al. 2008; Turner 2015), their morphotype, density
and porosity as well as the timing of their production. Strong meteorological events can also
perturbate C fluxes from the water column with an increasing time lag over depth (e.g. Pedrosa-
Pàmies et al. 2019). Admittedly, C budgets suffer from lack of time integration into the
analysis. Our study regarding PAP site is also concerned as it undergoes a substantial
seasonality (Cole et al. 2012; Giering et al. 2017). Although, we do not have enough
understanding of vertical time lag to change the model and to avoid such bias yet. Some long-
term observatories such as BATS in the Bermuda Atlantic or HOT in Hawaii provide
biogeochemical flux time series but monthly sampling focuses mostly on the euphotic zone
and does not investigate the mesopelagic zone enough. Sampling at discrete times following
the sink of a bloom (e.g. Le Moigne et al. 2016) could be a solution, which would nevertheless
entail a significant cruise planning effort.

## 4.4 Grounds for improvements
Anderson & Tang model allowed us to have a comprehensive vision of the remineralization
processes in the mesopelagic zone by including the interactions between various
compartments, completing *in situ* measurements with a comprehensive vision of the
mechanisms at stake. The described inversion of the Anderson & Tang model provided
meaningful estimations of the parameters of interest. However, as most models represent
complex phenomena, some processes are not fully and properly captured by the model. Below,
we provide a list of processes that may help refining mesopelagic C budget estimations.

### 4.4.1 Other microorganisms



Are not included in the model, the role of microbial eukaryotes, viruses, and the input of C by
chemolithotrophs whose potential role has gained in importance (Herndl and Reinthaler 2013;
Lara et al. 2017; Kuhlisch et al. 2021; Luo et al. 2022). For instance, eukaryotes can dominate
microbial biomass on bathypelagic particles (Bochdansky et al. 2017), and have the potential
to promote the aggregation of particles (Jain et al. 2005; Chang et al. 2014; Hamamoto and
Honda 2019; Xie et al. 2022). Viruses could be the main cause of prokaryotic and
phytoplanktonic mortality. Thus, DOC fluxes could be attributed to them, in particular with the
cell lyses they provoke (Fuhrman 2000 and ref within, Lara et al. 2017; Kuhlisch et al. 2021).
In the North Atlantic, 9 to 12% of cells could be infected by viruses which would cause a DOC
production of 0.1 mg C m$^{-3}$ d$^{-1}$ (Wilhem and Suttle 1999). For comparison, PHP results on PAP
before integration (with a conversion factor of 0.5 kg C mol$^{-1}$ Leu) were mostly below this
value. In addition, inorganic C fixation by chemoautotrophy would be of the same order of
magnitude as PHP$_{non-sinking}$ rates (Herndl et al. 2005; Reinthaler et al. 2010). It would be
important to verify what microbial eukaryotes, chemolithotrophs or viruses contributions are,
even if the poor understanding of these processes currently prevents properly integrating them
into models.

### 4.4.2 Lifestyles

More surprisingly, sinking prokaryotes are poorly considered as they are not sampled with the
Niskin bottles classically used in oceanography (Planquette and Sherrell 2012; Baumas et al.
2021). However, the use of the MSC at PAP DY032 allows us to access fractions of particulate
organic carbon that will allow us to evaluate the importance of sinking prokaryotes. We have
seen that their C demand is not negligible and represents 18% of total C demand. Anderson &
Tang model distinguishes sinking particles from neutrally buoyant particles, each with distinct
attached communities. Since sampling with MSC only allows us to separate what is sinking
from what is not, we merged free-living prokaryotes with those attached to neutrally buoyant
particles without distinction. However, unlike free-living prokaryotes, prokaryotes attached to
neutrally buoyant particles have access to POC and must produce enzyme activity with
different metabolisms than their free-living counterparts. On the other hand, prokaryotes
attached to neutrally buoyant particles are also different from prokaryotes attached to sinking
particles since they do not undergo changes in temperature and pressure related to the sink.
They must therefore surely have intrinsically different PGE and associated remineralization
rates. It would therefore be valuable to consider them as a third distinct group in laboratory
experiments and sampling. Contrary to the sinking or ascending particles which are naturally
split by their sinking/ascending velocity (e.g. respectively Smith et al. 1989; Cowen et al. 2001;



McDonnell et al. 2015), no means allow the selective and exclusive sampling of neutrally
buoyant particles. The only valid way is to use the MSC to let the sinking particles fall into the
lower compartments and to filter the "non-sinking" part to retain the particulate fraction.
However, it is known that filtration affects the activities of prokaryotes and generates biases
(Edgcomb et al. 2016). This makes investigations of prokaryotes associated with neutrally
buoyant particles particularly challenging and future endeavors should urgently attempt to
target them.

### 4.4.3 OC inputs


Continuing in the same line, the inputs of C that the model takes into account are only the
gravitational POC and the DOC. We chose to artificially increase the gravitational POC flux
to add sources of neutrally buoyant particles in the form of PIPs (eddy subduction pump,
metazoans migrations and large-scale physical pumps). Indeed, Boyd et al. (2019) clearly
showed that these PIPs can be of paramount importance (here we have estimated them at 51.6%
of the gravitational flux). Accounting for these neutrally buoyant particles through the POC
flux was performed due to the model structure. Yet, explicitly describing them in a dedicated
compartment of the model could be an improvement for future research, as these neutrally
buoyant particles have an effect on the whole system, including the prokaryotes linked to
various types of particles and their predators or on particle fragmentation. Given the existence
of the neutrally buoyant particle compartment, it is feasible to adapt the model to account for
these C inputs. This is even more relevant as new optical instruments have flourished (e.g.
Briggs et al. 2013; Giering et al. 2020; Picheral et al. 2022) and would make it easier to better
quantify these neutrally buoyant particle fluxes.

### 4.4.4 *In situ* pressure effect


Our last major concern deals with the fact that neither Niskin nor MSC avoid disruption
introduced through the process of depressurization when samples are collected at depth
(Tamburini et al. 2013; Garel et al. 2019). Heterotrophic activities associated to non-sinking
prokaryotes are known to decrease with depth but were mostly sampled without taking care of
the *in situ* pressure (e.g. Turley and Mackie 1994; Arístegui et al. 2009). From our knowledge,
some devices such as the $IODA_{6000}$ (Robert 2012) were specifically designed to measure *in situ*
PR of non-sinking prokaryotes. However, enigmatically high PR values (2-3 orders of
magnitude higher than PHP) are measured by $IODA_{6000}$, making it difficult to have confidence
in these *in situ* measured PR rates. During the PEACETIME cruise, we use a pressure-retaining



sampler (methods presented in supp data), allowing for the first time to access both PHP$_{non-}$
$_{sinking}$ and PR$_{non-sinking}$ rates and to compare it with classical depressurization procedures (Fig.
S1). We observed that activity rates of non-sinking prokaryotes kept under pressure were
always higher when kept at *in situ* hydrostatic pressure than their decompressed counterparts
and, surprisingly, seem to increase with depth rather than decrease typically depicted and found
when the samples are decompressed (Fig. S1). Focusing on PR$_{non-sinking}$ rates, obtained values
are also several orders of magnitude too high to be realistic in regard to C-Budgets and prevent
us from calculating PGEs. As PHP and PR are linked, it is very likely that the pressure effect
(here, an increase) is reflected on both and thus in the associated PGE$_{non-sinking}$. Taking
hydrostatic pressure into account could thus drastically affect C-budgets and even for
zooplankton respiration as we saw in the model that they are really sensitive to PGE$_{non-sinking}$.
We highly recommend using either direct *in situ* measurements or pressure retaining systems
for future research. This advice should be followed carefully, especially from 500m where the
pressure effect starts to be very important (Fig. S1), while the piezosphere was previously
considered below 1000m depth (Jannasch and Taylor 1984; Yayanos 1986). Furthermore, this
shows the crucial interest to measure points below 500m in order to get a global trend of the
profile, which could not have been done here for sinking prokaryotes (the MSC were deployed
only up to 500m during the cruise DY032). From a C-budget point of view, taking *in situ*
pressure into account will increase C demand of free-living prokaryotes well adapted to their
living depth.

The effect of pressure acts inversely on sinking prokaryotes, as they are surface prokaryotes
(unadapted to high-hydrostatic pressure) that undergo a dynamic pressure increase as the
particle sinks (Baumas et al. 2021; Tamburini et al. 2021). Besides, repeated results (Tamburini
et al. 2006, 2009, 2021; Riou et al. 2018) have shown that, while performing a sinking
simulation experiment the activities of sinking prokaryotes are affected during the sink. For
instance, they noticed that the aminopeptidase activity was always lower with increasing
pressure over time than at atmospheric pressure on diatom aggregates (Tamburini et al. 2006).
This may reflect the stress endured by the sinking prokaryotes as they experience the sink. This
could also be another explanation of why the fraction of hydrolyzed C released (**α**) tends to
decrease with depth as it is directly linked with aminopeptidase activity. In view of these
statements, it is not surprising that the PHP and PR, and therefore a PGE, are impacted by
increasing pressure (e.g. Stief et al. 2021; Tamburini et al. 2021). Only the RESPIRE from
Boyd et al. (2015) provides *in situ* measurements of the PR of sinking prokaryotes. However,
in line with the previous comments, it gives unrealistically rather high values. Thus, sinking



simulation experiments remains, in the present, the best alternative to understand the mechanics
of sinking prokaryotes during the sink of their associated particle. Several systems exist to
simulate the sink (e.g. de Jesus Mendes et al. 2007; Grossart and Gust 2009; Tamburini et al.
2009; Mendes and Thomsen 2012; Dong et al. 2018; Stief et al. 2021; Liu et al. 2022), all
showing a general tendency that hydrostatic pressure affects activities (and diversity) of
surface-originated prokaryotes, decreasing the integrated C-demand when taking into account.
Handling high-pressure sampling or experiments requires much more effort and material than
usual methods. However, it seems highly worthy when investigating both, sinking and non-
sinking prokaryotes activities, in regard to C-budget purposes.

## 730   5. Conclusion

By combining *in situ* data from the DY032 cruise at the PAP site with inversion of the
Anderson & Tang model which includes known processes from the biological C pump, we
provide robust and ecologically realistic estimates of key parameters and to better characterize
the patterns at stake.
1) We showed that the most sensitive parameters in the model are the ones related to

prokaryotes such as prokaryotic growth efficiencies, leucine-to-carbon conversion

factor, and C hydrolyzed by sinking prokaryotes released to the surrounding water.

2) By inversion of Anderson and Tang's model, we determined consistent values of the

parameters listed above.

3) We showed that using these values instead of the classical mean from literature or

inadequate theoretical values resulted in a more consistent and realistic C-budget than

previously considered.

4) Additional measurements are needed to better understand both prokaryotic growth

efficiencies and Leucine-to-Carbon conversion factors in the mesopelagic zone.

However, we recommend measuring fewer fluxes for which we are confident

associated with inversion model procedures in order to access parameter values

challenging to measure in other places, cruises, or seasons.



Fig. 2 summarizes processes involved in mesopelagic C budgets estimations and highlights
missing knowledges. We attempt to classify the processes according to their degree of
understanding (well known, insufficient data or unknown) and point out that majority of these
processes require a better understanding. Among others, it is crucial to quantify the roles of
microbial eukaryotes, viruses, and chemoautotrophs in the entire process of C budgets.
Suspended particles should have a dedicated well-identified compartment in future studies
instead of being neglected and drowned into others. Finally, accounting for *in situ* hydrostatic
pressure when studying prokaryotic C demand is key. This is because: 1) it may reduce PCD
for sinking prokaryotes unadapted to increasing pressure and 2) it may increase PCD for free-
living prokaryotes well-adapted to their living depth.

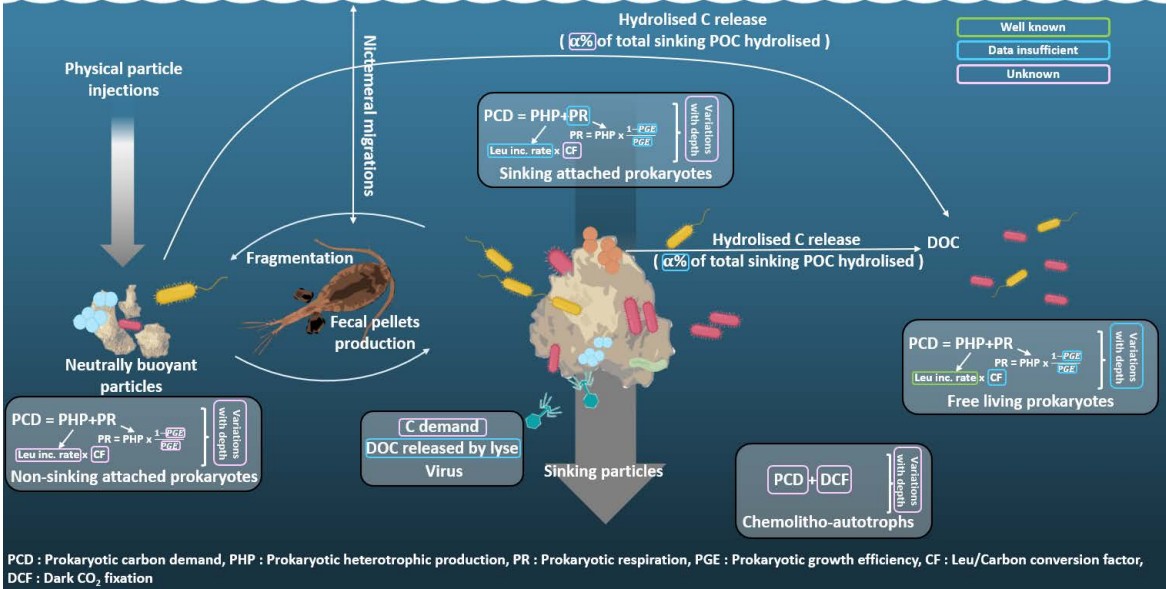

*Figure 2: Sinking particles export carbon (C) down to the mesopelagic zone through gravitational POC*
*fluxes where this latter is attenuated to satisfy C demand of different groups of organisms such as*
*prokaryotes living attached to sinking particles, attached to non-sinking particles, or free-living*
*prokaryotes. In turn, viruses and chemoautotrophs can increase the amount of usable labile C.*
*Quantifying C demand and role on POC fluxes of these different groups is crucial to truly assess C*
*sequestration in the deeper layer of the water column. However, a multitude of uncertainties remains*
*for each group. The quantities enclosed in green are well known, in blue lack data and in pink are*
*unknown. C demand is the sum of heterotrophic production (PHP) and respiration (PR). The*
*understanding of these two quantities is currently better for the free-living prokaryotes whereas data*
*are still insufficient for sinking prokaryotes and even absent for prokaryotes attached to non-sinking*
*particles. Moreover, to build C budgets, these variables are integrated over a few hundred meters of*
*water column and the relationship between in situ pressure and C demand remains often neglected even*
*if this relationship highly depends on the prokaryote type considered (not constant for sinking*



*prokaryotes unadapted to the increased pressure, constant for free-living prokaryotes well adapted to*
*their living depth and constant for prokaryotes attached to non-sinking particles which can be adapted*
*or not if the particle was sinking before being stopped in its sink ).*

# Code/Data availability

The codes and data to reproduce the results are available at
https://github.com/RobeeF/InverseCarbonBudgetEstim

# Author contribution

The idea was conceived by CB, CT and JCP. Sampling and experiments onboard PEACETIME
cruise were conducted by CT and MG. The data processing of PAP DY032 data was conducted
by CB with advices from FLM, and the one from PEACETIME data by CB and MG. RF
designed the inversion detection methodology and performed the estimation with advices from
LM. CB and RF led the writing with significant contributions from all authors.

# Acknowledgement

We thank the crew and officers of the R.R.S. DISCOVERY (NERC) for their help during the
PAP DY032 cruise and of the N/O Pourquoi Pas? during the PEACETIME cruise. This study
is a contribution to the PEACETIME project (MISTRALS CNRS INSU, doi:
10.17600/17000300) managed by Cécile Guieu (LOV) and Karine Desboeufs (LISA). We
warmly thank F. Van Wambeke, S. Guasco and B. Zancker for onboard works (enzymatic
activities, sugar and amino acids concentrations measurements) during PEACETIME cruise.
We wish to express our gratitude to F. Van Wambeke, S. Giering, T. Anderson and A. Belcher
for stimulating and informative discussions. This manuscript is a contribution of the APERO
project funded by the National Research Agency under the grant APERO [grant number ANR
ANR-21-CE01-0027] and by the French LEFE-Cyber program.

# Competing interests

The authors declare that they have no conflict of interest.



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
