# Peer review of "Reconstructing the ocean's mesopelagic zone"

_Biogeosciences, 2023_

## Author Response (AR1)

**Reconstructing the ocean's mesopelagic zone carbon budget: sensitivity and estimation of parameters associated with prokaryotic remineralization**

Chloé Baumas[1]*#, Robin Fuchs[1,2]*, Marc Garel[1], Jean-Christophe Poggiale[1], Laurent Memery[3], Frédéric A.C. Le Moigne[1,3], Christian Tamburini[1]

REVIEWER 1 :

**Our answers are in bold**

The authors present a relevant modeling exercise which may greatly contribute to reconcile carbon flux and demand in the mesopelagic ocean. The topic is of general interest and the approach original. The authors use inverse modeling to estimate critical parameters for the estimation of the carbon demand in the active mesopelagic zone. Overall, the manuscript is well written, and the methods are clearly presented. The conclusions are sound and may help to reconstruct the carbon budget in the mesopelagic zone.

**We thank the reviewer for positive comments and interest.**

Yet, there are some concerns that should be addressed. A major issue is the use of a fix CF for prokaryotic carbon demand. This factor has a major impact on PHP estimates (as the authors also conclude), and a better justification for fixing this parameter is needed. Apparently, the CF was not included in the sensitivity analysis using the Sobol indices. In table 2, the authors present the Sobol indices for 20 parameters, while in the text they state that they tested 24 parameters. Even if the authors can only optimize 4 parameters, it seems that CF would be more influential than the proportion of sinking POC taken by sinking prokaryotes, as the authors clearly state in the conclusions (lines 735-737).

**We agree that the CFs have a major impact on PHP and therefore on the resulting C-budget. As the CFs are not directly parameters of the model described by Anderson & Tang 2010, we could not properly include them in the sensitivity analysis. However, we still tried to add them in the inversion as shown in Table S4 (2 last columns). To make it more visible, we detailed the sentence Line 283 as follow:**

**"In this respect, the CFs have been fixed to 0.5 kg C mol Leu-1 (Estimates without fixing the CFs have however been carried out, see Table S4 in supp. data)."**

On the other hand, the discussion is extremely long (more than 10 pages), the authors should make an effort to reduce it. As a suggestion, part of the discussion could be moved to results and the section about grounds for improvements could be considerably cut.

**We agree and reduced the discussion especially the pressure part "4.4.4 In situ pressure effect" which was unbalanced regarding the other parts.**

Specific comments

-Lines 86-87. The authors could indicate that these boundaries would vary on space and time.

**For the sake of clarity, we changed the original sentence to:**

**"Indeed they specifically designed a method to determine from CTD-cast variables (fluorescence, O2 concentration, potential temperature, salinity, and density) accurate boundaries which vary in space and time ."**

-Line 89. What do the authors mean by "other response elements"? I guess they refer to other sources of discrepancy. Please, clarify.

**We understand that "other response elements" is too large and switched it by "other sources of discrepancy"**

-Line 99. Here and elsewhere in the text: I suggest changing "attached to sinking particles prokaryotic communities" to "particle-attached prokaryotic communities".

**We agree that changing this term as the reviewer suggests helps to be clearer.**

-Line 103. Change "values form the mean of" to "mean values from".

**We agree as well to changed according to reviewer requirement**

-Line 141. Change "of mesopelagic" to "the mesopelagic".

**We agree as well and thank the reviewer for pointing this and previous typos out.**

-Lines 147-151. Please clarify which data originate from cruises or from the literature.

**The data used are all coming from cruises but are already published in diverse papers as presented in the Material & method section and specifically summarized in Table 1.**

-Lines 201. I suggest changing "primarily" to "initially".

**We agree and changed in consequence**

-Line 205: Indicate the study area in Belcher et al (2016).

**We specified the sentence "particles obtained by Belcher et al. (2016) » by « particles obtained at the PAP site during the DY032 cruise by Belcher et al. (2016) »**

-Line 206. Clarify the meaning of "agg".

**We meant "per aggregate". Regarding the nomenclature used in this paper, we changed this unit by nmol particle-1d-1**

-Table 2. There are only 20 parameters in the table, while in the texts the authors refer to 24. Please, clarify.

**Indeed, this is a mistake. There are 20 parameters in the model and eventually 22 when we tried to add the CFs.**

-Lines 466-469. Please revise the sentence for clarity.

**The sentence is now phrased as follow : "As further comparison, the non-integrated data from DY0312 allows us to calculate a PGEsinking (using PGEsinking = PHPsinking/(PHPsinking+PRsinking) according to del Giorgio and Cole 1998). The result is thus, a depth-specific PGE instead of a depth-integrated PGE."**

-Lines 513-514. Please revise for English usage.

**We agree that this sentence is confusing and we've shortened it to :**

**Giering et al. (2014) results were mainly due to the difference in CF applied on their data, i.e. 0.44 kg C mol Leu-1.**

-Line 532. I suggest changing "under" to "below".

**We agree and changed in consequence**

-Lines 542-548). This is information is somehow repetitive.

-Line 572. I would say that PGE and CF may be related but not that they play similar roles.

**CF and PGE do not play similar roles at the biological level, but only in the mathematical calculation of the discrepancy, which is what we meant by "in the current formulation of the model." line 573.**

-Lines 616-617. Please revise for English usage.

**We changed the original sentence by:**

**"The role of microbial eukaryotes, viruses, and the input of C by chemolithotrophs (Herndl and Reinthaler 2013; Lara et al. 2017; Kuhlisch et al. 2021; Luo et al. 2022) are not included in the model."**

-Lines 665-666. Please revise for English usage.

**This sentence was indeed unclear and not necessary. We therefore deleted it as the two first sentences of the paragraph were sufficient.**

-Lines 735-737. This conclusion is incoherent with not optimizing CF, considering that is one of the most sensitive parameters.

**See next comment**

-Lines 738-739. This conclusion is not true, the authors do not use model inversion to optimize CF.

**Even if we tried to include CFs in the optimization procedure (Table S4), this is true that we did not perform the Sobol analysis and the main inversion of CFs. After reflexion, we removed "leucine-to-carbon conversion factor" from the sentence line 736**

**REVIEWER 2**

**Our answers are in bold**

This study by Baumas et al. builds on the recognition of the still large uncertainties about the biogeochemistry of the ocean´s mesopelagic zone and attempts to constrain them by applying the adaptation of Giering et al. (2004) of Anderson and Tang´s (2010) model and focusing on the role of prokaryotes. Consequently, their contribution is not really novel and they end up with a carbon budget very similar to that of Giering et al. (2014): less demand than input of mesopelagic organic carbon, thus allowing a certain amount of C being exported to greater depths. The only conspicuous difference between both studies (Fig. 1) is that, since they use an order of magnitude lower value for the growth efficiency of prokaryotes attached to sinking particles (PGEsinking), the respiration of these prokaryotes (PRsinking) becomes a relevant contribution. This improvement seems minor. Moreover, both Giering et al. (2014) and the present study´s better matches are strongly dependent on the choice of the Leu-to-C CF. Although their CF (0.5 Kg C mol Leu-1) is more accurate that the theoretical, unrealistically high value of 1.55 Kg C mol Leu-1), they both share the fact of an unrealistically fixed value for the entire depth layer, which is also the same for both sinking and non-sinking particles. One could easily argue that a few other possible combinations of CF and PGE values for sinking and non-sinking prokaryotes would end up with similar carbon budgets.

That said, I appreciate the authors´ effort of distinguishing between 3 fractions of prokaryotes rather than only 2: free-living and among the particle-attached, those attached to sinking and to neutrally buoyant (non-sinking particles). This is a conceptual improvement that, although lacking appropriate in situ measurements, suggests a way for future improvements.

**We thank the reviewer for positive comments and interest.**

I greatly miss the lack of any reference but tangential (when describing the PIP of Boyd et al. XXX) to the role of mesopelagic fish, either migrant or non-migrant, as sources of POC and DOC to the mesopelagic as well as contributors to remineralization through respiration. Micronekton is an important living biomass share in the mesopelagic zone present worldwide (Irigoien et al. 2014, Nat Commun) including the NE Atlantic (Peña et al., 2020, Mar Environ Res). It is surprising that this potentially important organic carbon pool and the associated fluxes is not considered. Similarly, when considering mesopelagic zooplankton they disregard the existence of diel vertical migration (DVM, Kelly et al. 2019, Front. Mar. Sci.). I wonder how their model would perform if these processes were included.

**The role of fish in the C budget of the mesopelagic zone, whether as input or remineralization, must certainly have an impact. Especially as a number of studies are beginning to emerge on this subject. Observations and process studies of their contributions to the ocean carbon cycle are lacking relative to phytoplankton, zooplankton and other taxa. Around 10 studies are listed for instance in "https://aslopubs.onlinelibrary.wiley.com/doi/full/10.1002/lno.11709". However the corresponding % of the POC fluxes is not necessarily provided (specially for the mesopelagic zone) and it is therefore hard to accurately set up a model with these values. Having said that, we agree that this is an area to explore for the future.**

As mentioned earlier, I consider that the authors' choice of fixed values as model inputs largely affect the estimation of parameters. The 3 mg C m-2 d-1 value for DOC export for instance (L183) seems to be arbitrarily low in view of the overlooking of the potential role of mesopelagic fish.

**These are not necessarily fixed values : we are using the model with discrete data measured during one specific field cruise. Unfortunately, we were unable to find any estimates of DOC fish excretion or production fluxes in the mesopelagic zone. The most similar processes are the DOC excretion or production fluxes of zooplankton which correspond to the 3 mg C m-2 d-1**

The discussion is too lengthy for just 2 figures, with one of them appearing after the conclusions and looking more like a graphical abstract. I suggest reconsidering its placement earlier in the discussion, as well as reducing the length of the discussion substantially.

**We agree that the pressure part "4.4.4 In situ pressure effect" is unbalanced regarding the other and significatively reduces it.**

L28. Delete the article ¨the¨ and use the singular for "respiraton".

**We thank the reviewer for noting these details**

L21. I understand the grounds for their later constraining of the "Active Mesopelagic Zone" to the 127-751 depth layer using the RUBALIZ method, but the mesopelagic zone is nor "roughly" located between 200 and 1000 m, that is its definition!

**It is true that the common definition of the mesopelagic zone is located between 200-1000m. However, it is also sometimes defined as "below 1% of the PAR (photosynthetically active radiation)", between 100-1000m or below the MLD for instance. That is why we added "roughly". On reflection, we changed "roughly" to "typically" in this sentence.**

L43-47. Confusing.

**We slightly modified the sentence to make it understandable. The sentence is now read as : "Five downward pathways of organic matter export to the mesopelagic zone are defined: phytoplankton (senescent cells, colonies, spores, cysts), zooplankton (carcasses or fecal pellets), aggregates (marine snow of different compositions including the two latter categories), vertical migration of zooplankton and mixing/diffusion/advection (Siegel et al. 2016; Le Moigne 2019)."**

L48. Please connect better the gravitational sinking with the 5 pathways from the above paragraph.

**As requested, we changed the sentence by : "Gravitational sinking POC supply, combining the 3 first pathways described above, constitutes the main organic carbon input to the mesopelagic zone (Boyd et al. 2019)."**

L54 and 63. No need for hyphen.

**The hyphens are now removed**

L64. And DOC released by DVM organisms.

**We believe this is out of context here, as we're only talking about degradation of particles in this paragraph.**

L68-69. Please re-write and provide more details. A reference to the POC-DOC continuum seems pertinent here.

**In line with the previous comment, we believe this is out of context here, as this paragraph is to introduce the degradation of particles by heterotrophs such as prokaryotes or zooplanktons**

L123-128. This is what it is all about. I follow the authors' argument but the CF and PGE are intrinsically variable and therefore a large source of uncertainty derives from the use of fixed values for the entire mesopelagic zone.

**Yes, this is why wisely choosing these parameters is crucial to determine the reconciliation or the imbalance of carbon budget.**

L138. "overlooked but widely used" is a paradox, please re-write.

**We agree and modified by "widely but inadequately used".**

L143-144. The authors do not really discuss the carbon sequestration by the BCP.

**We understand and removed the end of this sentence which is now :"3) to discuss our results in the context of mesopelagic carbon budget "**

L189. Insert a comma after "that is".

**A comma is now added, as requested**

Table 1. The large (order of magnitude) difference in PHP between sinking and non-sinking prokaryotes is not sufficiently supported.

**We do not understand the reviewers's comment. In Baumas et al 2021, we clearly show order of magnitude differences between both (as in Turley & Mackie 1994). Even if attached individual cells exhibit a higher PHP, in a given volume, particles are "rare" and free-living prokaryotes outnumber sinking prokaryotes. This explains the order-of-magnitude difference between the two when viewed per unit volume.**

I am sorry but I do not fully understand the concept of "identifiable model" (P10 and beyond).

**"Identifiable" is a mathematical term. To avoid confusion, we added the definition in the text as follow : "line 281: "To make the model identifiable (i.e. sufficiently constrained to estimate the true value of the parameters), the number of input ..."**

L300-301. Maybe it is only "and" missing between 0.27 and 0.25 but this sentence is difficult to follow. Consider re-writing it for clarity.

**We agree and splitted the sentence into two as follow : "Fluxes related to sinking prokaryotes, i.e. their PHP and their PR, appear to be highly influenced both by $\Psi$, $\alpha$, and PGEsinking. For instance, our analysis yields to indices of 0.22 and 0.23 for $\Psi$, 0.24 and 0.24 for $\alpha$ and 0.27, 0.25 for PGEsinking respectively."**

The parameters $\psi$ and $\alpha$ should be explained upon first appearance (P10), not on current L311-312.

**Yes we agree**

L392-394. The authors should try and argue this important outcome more in depth.

**The context of this paper is to identify limitations and not necessarily to find a solution. Specifically solving this problem could be the subject of another dedicated study.**

L397. The detritivores group is mentioned here for the first time and with a somewhat awkward notation (1-ψ). Please try and justify this better.

**The term is mentioned 8 times in the text (first introduced line 67). This group is defined as the zooplankton feeding on particles. The notation "(1-ψ)" is the one used in the model equations from Anderson & Tang 2010 paper and we would like to keep it for consistency purposes**

L399. "Zooplankton" or "zooplankters".

**We would like to keep the commonly use "Zooplankton"**

L407. And PR.

**The PR can be estimated from a PGE (as PR = PHP x (1-PGE)/PGE, del Giorgio and Cole 1998). Therefore, even if the appropriate PGE is often challenging to obtain, the PR is in theory not crucially required to build a C-budget. However, obtaining 2 independent ways to get measurements of PR and PHP would be a useful thing to have**

L409-415. This argumentation seems a bit tricky. The authors start by considering sugars (or carbohydrates polymers) as an important component of POC, but then they forget about then when using the results of experiments o aminoacid hydrolysis. Sugars are lost on their way. Can they elaborate more on this fact?

**We agree that this paragraph was confusing. We changed it by moving the sentence "Amino acids and sugar are major components of POC, constituting between 40 to 70% of POC in the mesopelagic zone (Wakeham et al. 1997)." a bit later after the sentence :"In the case of these two experiments, only the amino acids are considered and the experiments were conducted under laboratory-controlled settings."**

L447-448. I find this an interesting point, do they suggest a way of not having to measure PR in situ? I would appreciate, in the context for a reduction of the Discussion length, to see a full argumentation of this.

**Yes. Respiration is essential data for building a C budget, either directly or to have a PGE to convert PHP. However, PR is challenging to obtain. Indeed, a multitude of methods exist but lead to different results with sometimes few order of magnitude**

**differences. Regarding the model, the PR of free living prokaryotes matches the fraction of POC hydrolyzed by attached prokaryotes which is released in the surrounding water (i.e. α). As α seems more easily and reliably measurable, it could be another option to consider to assess PR of free living prokaryotes. For the sake of keeping the length of discussion we will not discuss this point further**

L484-. It looks like the use of 0.5 kg C mol Leu-1 would solve all our uncertainties and this is just the median of 15 values, not a value that comes from a direct measurement or some sort of empirical model relating CF to other variables. The authors should be more cautious about it.

**That is correct. That has been stated repeatedly (Line 97, 104, 486 & 491-501). We have also tried not to fix them (see Table S4). We are thus well aware and agree that this adds to the uncertainty. However, there exist no more measurements available to adjust our values.**

L488-489. Even for the non-sinking seems very optimistic that they have solved one of the many problems with the biogeochemical carbon fluxes in the mesopelagic zone.

L501. With what? Please explain briefly again here.

**This sentence was indeed unclear and we shortened it by : "Despite this, without having further data, we applied the same CF on sinking as the 0.5 recommended by Giering and Evans (2022) for non-sinking prokaryotes"**

L513-514. "from in situ data point of view"? Please re-write.

**We agree that this sentence is confusing and we've shortened it to : Giering et al. (2014) results were mainly due to the difference in CF applied on their data, i.e. 0.44 kg C mol Leu-1.**

L514. The acknowledgement of the huge role played by the choice of the CF is not a big advancement...

L518-519. To me, riverine aggregates are far from what the authors are considering in this paper.

**Yes we agree. However, this is what Giering et al. 2014 used to parameterize the model and, unfortunately, this is one of the only comparisons we have. This highlights the crucial need to improve the obtention of such data.**

L569-571. Two decimals are enough (3 are misleading). Please include also units.

**We thank the reviewer for the advice, this is now done**

L597-604. This is a key aspect that should be highlighted in an eventual reduction of the Discussion to focus on their model output.

**We agree and significatively reduces the discussion.**

L661-663. I think this is related to one of my major concerns but I do not follow the authors' argument. Please explain it better.

**Giering et al. 2014 or the model itself consider only the gravitational flux as input. Since then, other sources of C have been identified from the euphotic zone to the mesopelagic zone and can be remineralised as well and must be taken into account in C budget. As the model did not consider these other sources we artificially added an estimated proportion of POC input corresponding to PIPs from Boyd et al. 2019**

P23. There is a potentially huge problem in the fact that whenever the appropriate in situ pressure was used the activity of prokaryotes was ¨too high to be realistic¨ (L691) or gives "unrealistically rather high values" (L719). That is also related to the fact that the space devoted to in situ pressure effect is almost 2 pages long and just before the conclusion. I suggest re-sizing it and placing it earlier in this section.

**We agree that the pressure part "4.4.4 In situ pressure effect" is unbalanced regarding the other and reduces it.**